# Viral Haemorrhagic Septicemia Virus (VHSV) Isolated from Atlantic Herring, *Clupea harengus*, Causes Mortality in Bath Challenge on Juvenile Herring

**DOI:** 10.3390/v15010152

**Published:** 2023-01-04

**Authors:** Øivind Bergh, Torsten Snogdal Boutrup, Renate Johansen, Helle Frank Skall, Nina Sandlund, Niels Jørgen Olesen

**Affiliations:** 1Institute of Marine Research, P.O. Box 1870 Nordnes, 5817 Bergen, Norway; 2National Institute of Aquatic Resources, Section for Fish and Crustacean Diseases, Technical University of Denmark, 2800 Lyngby, Denmark; 3Norwegian Veterinary Institute, P.O. Box 64, 1431 Ås, Norway

**Keywords:** Atlantic herring, VHSV, challenge experiment, immunohistochemistry, rt-RT-PCR

## Abstract

Viral hemorrhagic septicaemia virus (VHSV) has been demonstrated to cause high mortalities in a wide range of teleosts, farmed as well as wild. In Europe, VHSV of genotypes Ib, Id, II, and III have been detected in wild fish, including Atlantic herring *Clupea harengus*, but disease outbreaks have not been observed in Atlantic herring and the effects on wild stocks are not well documented. Here, we have tested two VHSV isolates from herring (genotypes Ib and III, from the western coasts of Norway and Denmark, respectively) in a challenge experiment with herring (mean weight 2.59 g, SD 0.71 g) caught on the west coast of Denmark. The Norwegian genotype Ib isolate (NO-F-CH/2009) showed an accumulated mortality of 47% compared to 6% mortality with the Danish genotype III isolate 4p168 and zero in the unchallenged control group. In both groups, we found positive rt-RT-PCR and positive immunohistochemistry of VHSV from days 6 and 8 onward. With both isolates, the organs mainly affected were the heart and kidney. The results demonstrate the susceptibility of Atlantic herring to VHSV, and both genotypes gave pathological findings in several organs. Genotype III showed a low mortality rate, and the importance of this genotype for herring is therefore not determined. Genotype Ib showed both high prevalence and mortality, and this genotype is therefore likely to have a negative effect on wild Atlantic herring stocks. Further examinations to determine how VHSV can affect wild Atlantic herring stocks are needed.

## 1. Introduction

Viral hemorrhagic septicaemia (VHS), caused by VHS virus (VHSV) of several genotypes, is a disease known to cause high mortalities in many teleost species, both farmed and wild populations, having been isolated from approximately 80 different fish species in both marine and freshwater environments [1]. Disease outbreaks in Pacific herring *Clupea palassii* and several other wild fish species have been reported in the USA and Canada [2,3,4,5,6,7]. Susceptibility and high mortality rates have also been demonstrated experimentally in Pacific herring using North American VHSV isolates [8,9]. Based on long-term shedding from convalescing individuals with no clinical signs, Hershberger et al. [10] recently concluded that Pacific herring is a reservoir for the virus.

Using an ecological niche modeling approach to identify vulnerable disease-free regions, Escobar et al. [11] found VHSV in more than 140 fish species in marine and freshwater ecosystems, with a high diversity of lineages in Eurasia. Several authors [12,13,14,15] have demonstrated the presence of VHSV in various marine fish, also in Norwegian and Danish waters.

A screening of 1927 individuals from 39 different species of marine fish along the Norwegian coastal waters and fjord systems revealed positive detection of VHSV by means of real-time RT-PCR in twelve samples originating from Atlantic herring (*Clupea harengus*), haddock (*Melanogrammus aeglefinus*), whiting (*Merlangius merlangus*), and silvery pout (*Gadiculus argenteus*) [16]. All the fish tested positive in the gills, while four herring and one silvery pout also tested positive in the internal organs. Successful isolation in cell culture was only obtained from one pooled Atlantic herring sample. Sequencing revealed that the positive samples belonged to VHSV genotype 1b. All positive samples came from the northern county of Finnmark. The positive fish were asymptomatic carriers, and no fish with external symptoms of the disease were detected. This was the first detection of VHSV in Atlantic herring this far north, and the first detection in a silvery pout. Prevalence up to 16.7% of VHSV genotype Ib had earlier been detected in Atlantic herring and other wild marine fish in coastal areas around the English Channel, Eastern North Sea, Skagerrak, Kattegat, and the Baltic Sea; and the virus is assumed to be endemic in these waters [13,14].

Investigating Atlantic herring from the spring-spawning stock during the spawning season, Johansen et al. [17] found a prevalence of 19% in cell cultures of pooled samples from the brain, spleen, and kidney, and 33% with real-time RT-PCR from the same organs. Gills, which were analyzed only by real-time RT-PCR, displayed a prevalence of 69%. No disease symptoms were reported, and it could not be deduced from these data whether the high prevalence reflected an infection or only superficial detection of the virus in the mucus layer of the gills. The high prevalence of VHSV identified by Johansen et al. [17] (by both cell culture and rt-RT-PCR) would suggest that the virus may play a role in overall mortality on stocks of Atlantic herring, but a challenge experiment would be needed to further elucidate this.

Challenge trials on Pacific herring with VHSV isolates from the US (genotype IVa) resulted in high mortality rates [9]. Challenge trials with three stocks of Pacific herring showed that survivors from a primary challenge have low susceptibility to a secondary challenge with VHSV [18], indicating that fish might develop immunity. A low prevalence could indicate either a low exposure rate or immunity due to previous exposure. These studies did not reveal inter-stock differences in the cumulative mortality of Pacific herring due to exposure to VHSV. Earlier studies have hypothesized that the high prevalence of VHSV could be related to collapses in Pacific herring populations.

Considering the Atlantic herring in the North Sea, there were many reasons for the collapses in the 1960s, including overfishing [19], but clearly, all factors affecting stock dynamics and natural mortality should be elucidated. Thus, as VHSV virulence in Pacific herring has been established [9], as has the abundance of VHSV in Atlantic herring stock [13,14,17], the present study was designed as a challenge experiment in order to test the virulence of VHSV isolates of genotypes Ib and III, isolated from Atlantic herring in Norwegian and Danish waters, respectively, on juvenile Atlantic herring. 

## 2. Materials and Methods

### 2.1. Challenge Material

Two isolates of VHSV were used: DK-4p168, a genotype III isolate from Atlantic herring caught in Skagerrak waters outside of Denmark [15], and NO-F-CH/2009, a genotype Ib isolate from Atlantic herring caught in Finnmark waters in northern Norway [16]. Both isolates used were of low passage numbers (maximum 5 passages) and were propagated and titrated on BF-2 cells according to the standard procedures described by Lorenzen et al. [20].

### 2.2. Fish and Experimental Design

The animal experiments were carried out in Denmark, under Danish law, according to permission 2007 (561-1312, document BES40178 to Professor Niels Jørgen Olesen, Section for Fish Diseases, National Veterinary Institute, Technical University of Denmark). By this authorization, the experiments were judged, by the relevant national authority, to be in accordance with the WMA Declaration of Helsinki.

Atlantic herring (mean weight 2.59 g, SD 0.71 g; mean length 7.65 cm, SD 0.35 cm) were obtained from the harbor of Esbjerg, on the Danish west coast (North Sea), in collaboration with Fiskeri-og Søfartsmuseet (a local public aquarium). The fish were transferred to the National Veterinary Institute in Århus, Denmark, conditioned to the laboratory facility and tested for VHSV prior to the experiment. Thirty fish were screened (brain, heart, spleen, and kidney tissue) by the rt-RT-PCR protocol described earlier [21,22] and found negative. Initially, some mortality occurred that was investigated. The bacterium *Pseudomonas anguilliseptica* was isolated in pure culture from all examined fish. The fish were treated with florfenicol (Nuflor 300 mg/mL, Intervet) in the tank water at 10 mg/L for 5 days. After this initial problem, the fish acclimatized well, as no more mortality occurred and no other problems were observed before the beginning of the trial.

The experiment was carried out in 7 rectangular tanks covered with lids. (Figure 1a,b). The tanks were made specifically for this trial. Before the trial, the tank design was tested for its ability to support the fish for a prolonged time. Initially, the fish were stressed in the see-through tanks; however, a few simple wrappings with tape made the fish relax, and at the same time, observations into the tanks could be performed. After the adjustments, the tank design supported the fish well, showing free and natural behavior, including being curious and eating feed pellets with a good appetite.

Two of the three tanks for each virus isolate were used for mortality registration, with 26 fish in each and 30 L water, whereas the third was used for sampling during the experiment, with 60 fish (50 L water). The 7th tank was used as a non-infected control. The inoculum was prepared by the standard methods described by Lorenzen et al. [18]. On day 0 of the trial, the fish were infected by bath challenge for two hours with approximately 10^5^ TCID_50_ VHSV/mL water of one of the two virus strains, and thereafter approximately 50% of the water was changed. The control group was treated identically to the challenged groups and exposed to a mock challenge with supernatant from an uninfected cell culture.

The fish were kept in seawater of 30 ppt salinity. The seawater was obtained from Skagerrak outside Hirtshals and pumped through a pipe placed below the sand. The water was transported to the laboratory and kept in a tank with circulation until use. The water temperature was set at 10.5 °C, with acceptable short-term (from day to day) outer limits at +/−1.5 °C during the trial.

Feeding was conducted daily with commercial rainbow trout feed (INICIO+ 1.1 mm, from Biomar, Brande, Denmark). Feces, excess food, and other detrimental material were retrieved with a net daily, and a 50% water change was performed. 

### 2.3. Tissue Sampling

Five fish from each of the groups were sampled randomly on day 1 (post-infection/PI), day 2 PI, day 3 PI, day 5 PI, day 7 PI, day 11 PI, day 21 PI, and at the end of the trial on day 29 PI, for histopathology. On days 7, 21, and 29, an additional 5 fish from each virus group plus 2 fish from the negative control tank were randomly sampled for virological examination. Fish that were euthanized because of severe clinical disease and dead fish outside the pre-defined sampling dates were also sampled for virological examination. 

Originally, the sampling day was set to day 14 and not day 11, but unfortunately, an accident occurred, stopping the airflow to the tanks, resulting in several dead fish in each tank, as shown in the spike in mortality between day 10 and day 11 (Figure 2A). Several of the fish were found to be recently dead (by examining gills, eyes, and presence of rigor), and due to the limited amounts of fish, it was decided to use these fish as mid-trial samples and not perform the planned sampling on day 14.

For histopathology, fish were fixed in 4% neutral buffered formaldehyde after opening the abdominal cavity. The samples were placed at 4 °C, and at the end of the experiment, they were transferred to the National Veterinary Institute in Oslo for further processing. Formalin-fixed tissue samples were processed and embedded in paraffin wax. Sections (4 to 6 µm) were stained with hematoxylin and eosin and examined by light microscopy.

Immunohistochemical examinations were performed according to [22]. The monoclonal antibody (Mab) IP5B11 against the VHSV nucleoprotein [18] was used as the primary antibody, and biotinylated rabbit anti-mouse immunoglobulin (Dako) as the secondary antibody before incubation with the streptavidin-alkaline-phosphatase complex (Amersham Biosciences, Little Chalfont, Buckinghamshire, UK). The primary antibody was visualized by adding Fast Red salt (Sigma-Aldrich, Merck, Søborg, Denmark) to the Fast Red substrate solution.

### 2.4. Detection of Virus

The samples were tested for the presence of VHSV by real-time RT-PCR, by the protocol described by Jonstrup et al. [22], or by cell culture to show that it was possible to reisolate live virus. Cell culture isolation was conducted according to standard methods described in the OIE aquatic manual [23].

### 2.5. Bacteriology

Kidney swabs for virological examination were also subjected to bacteriological examination on blood agar (nutrient agar, Oxoid) and blood agar supplemented with 2% (*w/vol*) NaCl as described by Pedersen et al. [22].

## 3. Results

### 3.1. Clinical Disease and Mortality

Mortality increased from day 6 post-challenge, in particular in the group challenged with the genotype Ib isolate (Figure 2A,B). On day 18, it reached 47%. After that, no mortality was recorded in the experiment. In the group challenged with the genotype III isolate 4p168, mortality started on day 7 and reached 8% on day 9, with no further mortality observed throughout the experiment. No mortality was recorded in the unchallenged control group.

Prior to the onset of mortality, clinical signs were observed in the fish in both groups. First with the darkening of single fish (Figure 3A), often with a slight reddening of translucent parts of the head. This developed into more severe symptoms, with exophthalmia and hemorrhaging in the eyes; characteristic hemorrhaging was observed in the eyes and around the mouth and gill covers (Figure 3B).

Upon termination of the trial, some dark fish which didn’t develop more advanced symptoms, seemed to have recovered or in the process, with less pronounced darkening.

Upon sampling fish with external symptoms, hemorrhaging could be observed in the musculature when the skin was removed (Figure 3C).

No bacteria were recovered from the fish following the challenge.

### 3.2. rt-RT-PCR

Positive detection was found at all samplings from day 7 onward in both groups. However, whereas 4 of 5 individuals were positive in the group challenged with VHSV genotype Ib NO-F-CH/2009, only one of five individuals in the group challenged with VHSV genotype III DK-4p168 tested positive on day 7. On the two other sampling dates, days 21 and 30, slightly fewer individuals tested positive in the group challenged with VHSV genotype III DK-4p168 compared to the group challenged with VHSV genotype Ib NO-F-CH/2009, in which all individuals tested positive. It should be noted, though, that the sampling sizes were slightly different. However, most fish tested positive in both groups at the late sampling points (Table 1). All the fish that died or were euthanized due to clinical symptoms tested positive.

On days 1 to 5 and 11, only gills were sampled; on days 1 to 5, these tested negative for both virus isolates. On day 11, fish in relation to the accident, 9 out of 10 fish were positive in the gill sampled for genotype Ib NO-F-CH/2009, whereas only 2 out of 10 fish were positive for genotype III DK-4p168.

### 3.3. Cultivation of VHSV

The virus was cultivated in both experimental groups from herring sampled as dead or euthanized during the trial and in the late random samplings of fish on days 21 and 30. Thus, the results from the virus cultivation support the results from the rt-RT-PCR tests.

### 3.4. Histology and Immunohistochemistry

Pathology dominated by necrosis was seen in the heart, kidney, pancreas, intestine, and liver (Figure 4). The presence of the VHSV in the affected areas of both groups was confirmed by immunohistochemistry (Figure 3B–E). The organs most affected were the kidney and ventricle of the heart. No positive staining of the brain, gills, spleen, or intestinal wall was observed.

## 4. Discussion

This study demonstrates that Atlantic herring are highly susceptible to waterborne exposure of VHSV genotype Ib and, to some extent, genotype III, with pathological findings and virus detection in both challenged groups and mortality rates of 47% and 6%, in contrast to zero mortality in the unchallenged group. To the author’s knowledge, this is the first published challenge experiment with VHSV on Atlantic herring. The prevalence of VHSV has been studied in Danish [13] and Norwegian [12,16] waters, showing that Atlantic herring can be exposed to these viruses in their natural environments. Most of these isolations were from asymptomatic carriers.

Since the early 1990s, severe VHS epizootics with high mortalities have been observed in Pacific herring on the West coast of Canada and Alaska [2]. Kocan et al. [8] fulfilled Koch’s postulates by infecting SPF (Specific Pathogen-Free)-reared Pacific herring with North American VHSV isolates by bath challenge. It was found that these isolates were highly pathogenic by the immersion route, with mortality approaching 100%. Mortality began 4–6 days PI, and moribund fish displayed petechial hemorrhages on the lower jaw, isthmus, and eyes. Histopathological examination of tissues from moribund fish revealed multifocal coagulative necrosis of hepatocytes, diffuse necrosis of interstitial hematopoietic tissues in the kidney, diffuse necrosis of the spleen, epidermis, and subcutis, and occasional necrosis of pancreatic acinar cells. Experimental work and findings from natural outbreaks show that the North American VHSV strain is highly pathogenic for Pacific herring.

The first isolations of VHSV from Atlantic herring were in 1996 [15], and numerous isolations have been reported since then. However, none of these were associated with clinical signs of VHS, and no known mass mortalities of Atlantic herring have been associated with VHSV infection. As VHS is known to cause more severe infections and mortalities in juvenile fish, and as possible severe mortalities in juvenile herring would likely remain undetected under natural conditions, experimental infection has long been desired in order to assess the virulence and possible impact of VHS on Atlantic herring.

As no SPF Atlantic herring is presently available, the small herring included in this study were wild-caught and examined for pathogens before the onset of the trials. It should be emphasized that there were no findings of VHSV. Atlantic herring is not available in aquaculture, and there is limited experience in rearing the species for experimental purposes.

The large majority of VHSV isolates from Atlantic herring are of genotype Ib, which is the most prevalent genotype in the Baltic Sea, Kattegat, Skagerrak, and along the Norwegian West Coast [17]. Genotype III is predominantly present around the British Isles, with only very few isolations in Atlantic herring. It was therefore interesting to observe that genotype Ib caused significantly higher mortality in the herring than genotype III.

For both genotypes, however, the virus was still present in 78% of the surviving fish 30 days PI, indicating a strong ability of both genotypes to establish and persist in latent carrier fish.

The clinical and pathological findings in infected Atlantic herring with darkening, exophthalmia, and hemorrhages around the mouth and in the musculature, were very close to the pathology observed in Pacific herring [18] and in other species [14]. This also goes for the histopathological findings with necrosis of cells in internal organs primarily affecting heart, kidney, and pancreas.

The relatively high mortality in the experiment with the genotype Ib isolate, combined with the findings by Johansen et al. [17] that these viruses were present in relatively large amounts in Atlantic herring post-spawning at the Norwegian west coast, makes it possible to speculate that the virus may play a role in mortality in wild populations. Johansen et al. [17] showed a high prevalence of virus in the gills compared to internal organs and discussed whether this was because of exogenous virus particles attached to this external organ or whether the gills were important in the pathogenesis of the infection. In the present study, we were not able to demonstrate that the gills played a significant role in the early pathogenesis, and the gills only became virus positive at the same time as general viremia.

However, the results demonstrate that Atlantic herring is susceptible for VHSV and combined with the high prevalence found earlier [16,17], they support the hypothesis that these viruses may contribute to the mortality of wild populations of Atlantic herring, as also demonstrated for Pacific herring [10].

## Figures and Tables

**Figure 1 viruses-15-00152-f001:**
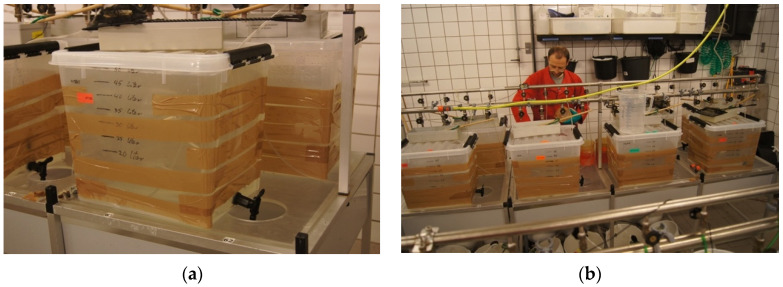
(**a**,**b**). The experimental system with fish tanks, of which 7 were used in the experiment.

**Figure 2 viruses-15-00152-f002:**
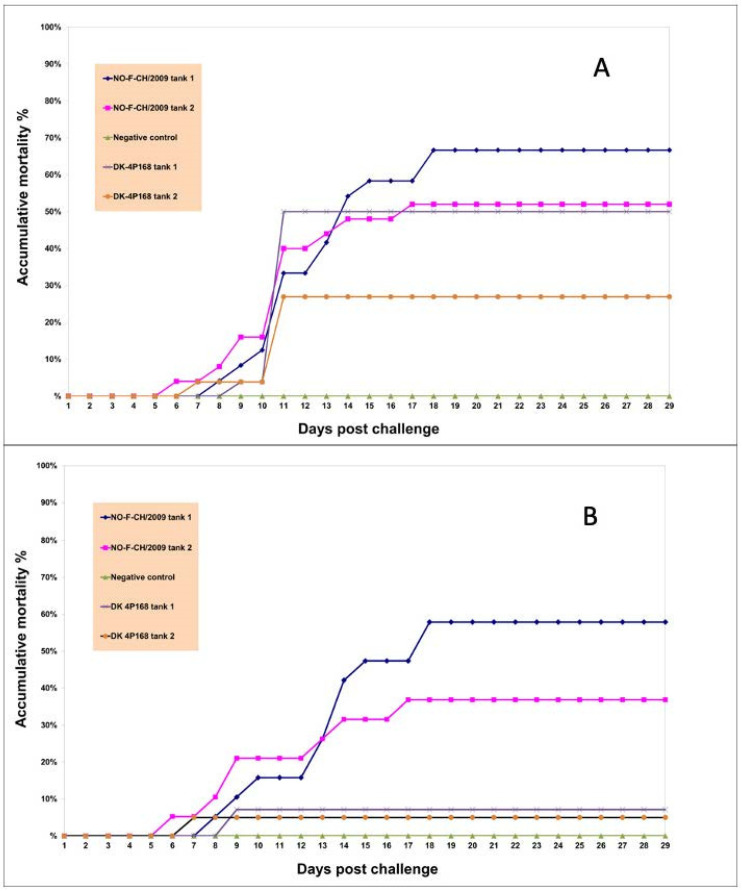
(**A**,**B**). Mortality in the three experimental groups of Atlantic herring, *Clupea harengus*, challenged with the VHSV genotype Ib isolate NO-F-CH/2009, the VHSV genotype III isolate 4p168, and the unchallenged control. In (**A**), all raw data are used, including mortality induced by the accident at day 11. In (**B**), the fish lost by the accident are subtracted, displaying only the mortality that could be related to the challenge.

**Figure 3 viruses-15-00152-f003:**
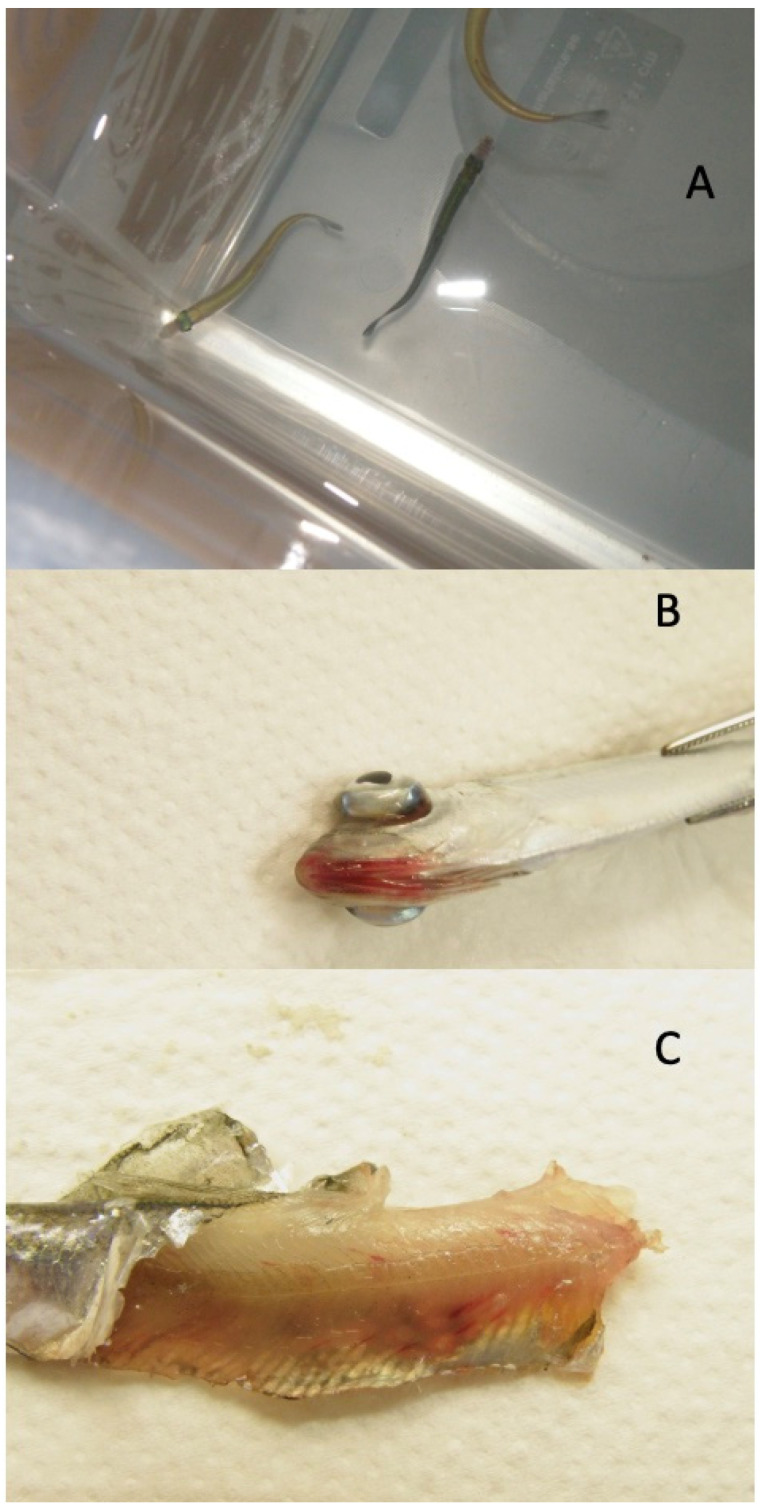
Gross pathology, *Clupea harengus.* Darkening of a single fish (**A**), often with a slight reddening of translucent parts of the head. Exophthalmia and hemorrhaging in the eyes, characteristic hemorrhages were observed in the eyes and around the mouth and gill covers (**B**). Hemorrhages in musculature (**C**).

**Figure 4 viruses-15-00152-f004:**
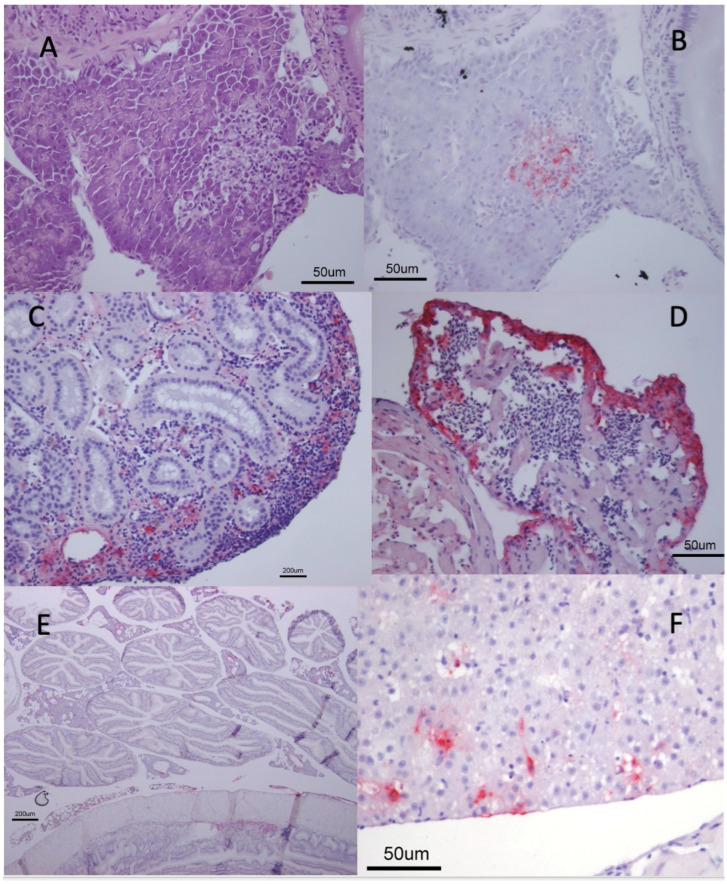
Haematoxylin staining (**A**) and immunohistochemistry (IHC) (**B**–**F**) of tissue samples from Atlantic herring challenged with VHSV genotype III 4p168 (**A**–**C**) or the VHSV genotype Ib isolate NO F-CH/2009 (**D**–**F**). (**A**) Necrosis in pancreas. (**B**) IHC of the same tissue as in (**A**), showing positive immunostaining in the area where necrosis is seen. (**C**) Kidney of the same individual as in (**A**,**B**) shows positive immunohistochemistry in the hematopoietic tissue. (**D**) Heart, showing positive IHC in the outer rim of the atrium and in lumen of the ventricle. (**E**) Overview of the abdominal cavity, showing intestine, pancreas, and pylorus. Positive staining in pancreas. (**F**). Positive IHC of hepatocytes in liver. The larvae were sampled at day 7 PI (**A**–**C**), day 9 PI (**D**,**E**) and day 14 PI (**F**).

**Table 1 viruses-15-00152-t001:** Number of VHSV positive samples by rt-RT-PCR (and cell culture on day 21 and day 30) out of the number of examined Atlantic herring, *Clupea harengus*, challenged with VHSV genotype III DK-4p168 and VHSV genotype Ib NO-F-CH/2009, respectively. On days marked with an asterisk (*), only the gills were sampled, whereas on other days, internal organs as well as the gills were sampled.

Days Post Challenge	VHSV DK-4p168 (Genotype III)	VHSV NO-F-CH/2009 (Genotype Ib)
1 *	0/5	0/5
5 *	0/5	0/5
7	1/5	4/5
11 *	2/10	9/10
21 rt-RT-PCR 21 Cell culture	3/5 (2/5)	4/4 4/4
30 rt-RT-PCR 30 Cell culture	8/9 7/9	8/8 7/9

## Data Availability

All data can be made available upon request.

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
