# Peer review of "Viral Haemorrhagic Septicemia Virus (VHSV) Isolated from Atlantic Herring, Clupea harengus, Causes Mortality in Bath Challenge on Juvenile Herring"

_viruses, 2023, doi:10.3390/v15010152_

Round 1

Reviewer 2 Report

Manuscript #2034997 (Viruses), “Viral Haemorrhagic Septicemia Virus (VHSV) isolated from Atlantic herring, Clupea harengus, causes mortality in bath challenge on juvenile herring” describes laboratory studies intended to assess the virulence of two VHSV genotypes (Ib and III) to Atlantic herring. With the documented impacts of VHSV genogroup IVa on Pacific herring, the question of Atlantic herring susceptibility to endemic genotypes has been uninvestigated and has remained outstanding for many years.  I am delighted that the authors made an effort to investigate this host / pathogen relationship.  Additionally, the authors should be commended for designing a very nice study and for describing their effort with a very succinct writing style.   Demonstration of elevated virulence of Genogroup Ib relative to Genogroup III in Atlantic herring is novel and worthy of publication.  My comments are rather trivial and amount to minor expansion of a few details that may enhance the strength of the paper.

There are a few unfortunate details that would have made the paper stronger; however, some of these weaknesses must be expected in any experiment; none represent fatal flaws that would change the outcome of the study. For example:

 - Experimental design: Only 3 tanks were used for each virus type: 2 for mortality assessments, and 1 for subsampling.  It would have been nice to include an additional mortality tank so that mean mortality and SD could have been reported.  Additionally, only a single (un-replicated) control tank was included.  Again, these are only unfortunate design limitations that I’m sure were based on logistical constraints.  No author response needed.

- The Day 11 mortality due to lack of tank aeration was unfortunate; however, these issues do occur periodically, and the authors did a nice job of documenting the event. No author response needed.

- The lack of availability of SPF Atlantic herring for experimental animals was unfortunate and required the authors to use captive wild herring as a source of experimental animals.  This could use a bit more explanation in the manuscript.  Although the experimental animals were screened for VHSV and found to be free of the virus, it is possible that herring survived prior exposure to genogroup III and therefore developed immunity and were refractory to subsequent laboratory exposure.  The authors should acknowledge or explore this possibility in the paper.  This prior exposure scenario certainly would not have influenced their susceptibility results to Genogroup Ib, because the fish demonstrated high susceptibility to disease and mortality.  However, both the test animals and the virus isolate from Genogroup III were from a sympatric location (Denmark), supporting the possibility that mortality did not occur because of prior exposure that resulted in adaptive immunity to this virus type.  This prior exposure scenario could possibly be refuted if the authors could document (from the literature) that exposure to Genogroup III provides cross protection to Genogroup IB.  If this were true, then we could assume that prior exposure to Genogroup III never occurred in their test animals because high susceptibility to disease and mortality from Genogroup IB persisted in this study.  Either way, the manuscript would benefit from some type of acknowledgement of the shortcomings associated with experimental animals with unknown exposure histories (possibly by expanding the paragraph at line 287).

A few other minor considerations for the authors:

- Line 87: consider eliminating this sentence.

- Lines 113-114: please state what tissues were screened (presumably the gills)

- Consider eliminating Figure 1A&B.  It is not necessary.

- Line 138: 9-12 C is quite a broad temperature range.  This range of temperatures in Pacific herring can result in dramatic differences in mortality for genogroup IVa.

 - Lines 141-144: (The negative control…”)  Consider eliminating these sentences; they describe standard BSL-2 procedures that can be assumed.

- Figure 2: Please label the legends with the virus strain, not the isolate #.  It is difficult for the reader to cross reference these isolate numbers.

Line 203 (and throughout) consider replacing “bleedings” with “haemorrhaging”.

Lines 237-242: Please expand Table 1 to include the virus cultivation results in addition to the RT-PCR results.  Also, there is a typo in the Table legend (rt-RT-PCR).

Line 262: consider replacing “non-invasive” with “waterborne.

- Line 279: replace “symptoms” with “signs”.

Again, this is a very straight forward article, and the authors did a nice job with the write-up.
